# Characterization and Antimicrobial Susceptibility Patterns of *Enterococcus* Species Isolated from Nosocomial Infections in a Saudi Tertiary Care Hospital over a Ten-Year Period (2012–2021)

**DOI:** 10.3390/diagnostics14111190

**Published:** 2024-06-05

**Authors:** Ali Al Bshabshe, Abdullah Algarni, Yahya Shabi, Abdulrahman Alwahhabi, Mohammed Asiri, Ahmed Alasmari, Adil Alshehry, Wesam F. Mousa, Nashwa Noreldin

**Affiliations:** 1Department of Medicine, College of Medicine, King Khalid University, Abha 61421, Saudi Arabia; prof071@hotmail.com; 2Department of Family Medicine, Aseer Central Hospital, Abha 62523, Saudi Arabia; abaid1406@gmail.com; 3Department of Microbiology and Clinical Parasitology, College of Medicine, King Khalid University, Abha 61421, Saudi Arabia; shabi.yahya@gmail.com; 4Department of Internal Medicine, Aseer Central Hospital, Abha 62523, Saudi Arabia; alwahhabi94@gmail.com (A.A.); mshweel666@gmail.com (M.A.);; 5Department of Anesthesia and ICU, College of Medicine, Tanta University, Tanta 31512, Egypt; 6Department of Internal Medicine, College of Medicine, Tanta University, Tanta 31512, Egypt; nashwanor@hotmail.com

**Keywords:** *Enterococci*, vancomycin, antimicrobials, nosocomial infections, prevalence

## Abstract

Introduction: The Enterococcus genus is a common cause of nosocomial infections, with vancomycin-resistant enterococci (VRE) posing a significant treatment challenge. Method: This retrospective study, spanning ten years (2012 to 2021), analyzes antimicrobial susceptibility patterns of *Enterococcus* species from clinical samples in a Saudi Arabian tertiary care hospital. Result: A total of 1034 Enterococcus isolates were collected, 729 from general wards and 305 from intensive care unit (ICU) patients. VRE accounted for 15.9% of isolates. *E. faecalis* was the most common species (54.3% of isolates and 2.7% of VRE), followed by *E. faecium* (33.6% of isolates and 41.2% of VRE). *E. faecium* exhibited the highest resistance to ciprofloxacin (84.1%), ampicillin (81.6%), and rifampicin (80%), with daptomycin (0.6%) and linezolid (3.1%) showing the lowest resistance. In *E. faecalis*, ciprofloxacin resistance was highest (59.7%), followed by rifampicin (20.1%) and ampicillin (11.8%). Daptomycin (0%), linezolid (1.5%), and vancomycin (2.7%) had the lowest resistance. VRE cases had higher mortality rates compared to vancomycin-sensitive enterococci (VSE). Conclusion: Eight different strains of *Enterocci* were identified. *E. faecalis* was the most commonly identified strain, while *E. faecium* had the highest percentage of VRE. VRE cases had a significantly higher mortality rate than VSE cases.

## 1. Introduction

*Enterococci* are gram-positive bacteria typically found in the human gut in short chains and pairs. They are of great concern worldwide due to their nosocomial nature and emerging drug resistance patterns [1,2]. Several *Enterococcus* species have been identified; among them, *Enterococcus faecalis (E. faecalis)* is the most commonly isolated species, accounting for 80–90% of nosocomial enterococcal infections. *E. faecium* accounts for 10–15% of these enterococcal infections [1,2,3]. The most common infections caused by *Enterococcus* species are lower urinary tract infections, such as prostatitis, cystitis, and epididymitis. Other infections caused by enterococci include bacteremia; catheter-related infections; intra-abdominal, pelvic, and soft tissue infections; wound infections; endocarditis; and respiratory tract infections.

Patients in the ICU are at a high risk of nosocomial infections due to lowered immunity, indwelling catheterization, and increased antimicrobial use. This group is at high risk from penicillin-resistant VRE, with limited treatment options [4].

Many antimicrobial agents, including cephalosporins, aminoglycosides, clindamycin, and semisynthetic penicillinase-stable penicillins, are intrinsically resistant to enterococci. *Enterococci* can also develop antimicrobial resistance through genetic mutations, further limiting treatment options [5]. Ampicillin is a convenient option for managing infections in penicillin-susceptible VRE. Reserved antimicrobials such as linezolid and daptomycin are used in life-threatening penicillin-resistant VRE cases, but tigecycline and quinupristin/dalfopristin should be individually evaluated before administration [5,6]. Antimicrobial resistance has increased significantly in the last decade, particularly since the COVID-19 pandemic [7]. The rising trend in antimicrobial resistance during the pandemic could be attributed to poor infection control practices and antimicrobial overuse [8,9]. 

This retrospective study was carried out in a tertiary care hospital in Saudi Arabia on clinical isolates of nosocomial infections from 2012 to 2021 to characterize and discover the antimicrobial susceptibility patterns of *Enterococcus* species; we think this will address a common problem in the world of hospital-acquired infection, especially among our national health care professionals.

## 2. Materials and Methods

This retrospective study was conducted in a tertiary care hospital over a 10-year period (2012–2021) to characterize *Enterococcus* species and evaluate resistance patterns. The hospital information system was used to collect data on 1034 isolates, with 729 from general wards and 305 from the ICU. We included in this data set all adult patients aged 18 years and above. Along with the requirements that the samples were from nosocomial infections and that the infections had been identified to be hospital-acquired (nosocomial), based on the fact that these infections are usually acquired after hospitalization and manifest 48 h after admission to the hospital, we also excluded any samples collected within 48 h of hospital admission. Thus, we excluded any samples from the pediatric population and any infections from community-acquired sources. The isolates included samples from abscesses, body fluids, blood, CSF, respiratory samples, soft tissue, urine, and wounds. All samples were collected in a sterile manner and sent to the microbiology laboratory. Samples were processed on defined culture media plates, including blood, chocolate, MacConkey, and cysteine lactose electrolyte-deficient (CLED) media, according to the standard operating procedure (SOP) of the microbiology laboratory, which was the Clinical and Laboratory Standards Institute (CLSI). Culture plates were incubated aerobically at 35–37 °C for 24–48 h. Gram staining was performed on samples from abscesses, body fluids, blood, CSF, respiratory samples, soft tissue, and wounds. Culture media plates were observed after 24 h for the presence of pathogenic isolates, their colony morphology, and Gram staining results. In the case of non-significant growth, the culture media plates were re-incubated for an additional 24 h. *Enterococcus* species were identified by Gram stain results as gram-positive cocci in pairs or short chains, morphology on culture media plates, and biochemical tests such as bile esculin and 6.5% salt broth.

Antimicrobial sensitivity was assessed for all positive isolates using the automated VITEK system, and antibiotic selection was performed according to the Clinical and Laboratory Standards Institute (CLSI) guidelines. The results were interpreted according to CLSI guidelines.

### Statistical Analysis

All data were entered and evaluated using Microsoft Excel 360 and Stata version 17 using Fisher’s exact test and a binomial generalized linear model for trends.

## 3. Results

### 3.1. Characteristics of Study Population

This study included 1034 *Enterococcus* species isolates: 647 (62.6%) from men, with 88 (53.7%) VRE and 559 (64.3%) VSE, and 387 (37.4%) from women, with 76 (46.3%) VRE and 311 (35.7%) VSE (*p* = 0.001). Of the isolates, 729 (70.5%) were from hospital wards, with 88 (53.6%) VRE and 641 (73.7%) VSE, while 305 (29.5%) were from the ICU, with 76 (46.4%) VRE and 229 (26.3%) VSE (*p* = 0.001). Overall, there were 164 (15.9%) VRE and 870 (84.1%) VSE (*p* = 0.01). Of the isolates, 561 (54.3%) were *E. faecalis*, with 15 (2.7%) VRE and 546 (97.3%) VSE; 347 (33.5%) were *E. faecium*, with 143 (41.2%) VRE and 204 (58.8%) VSE; and 126 (12.2%) were from other species, with 6 (4.8%) VRE and 120 (95.2%) VSE (*p* = 0.001) (Figure 1).

We used the following cultural sources: Abscess 60 (5.8%): 8 VRE (4.9%) and 52 VSE (6%); ascitic fluid 21 (2%): 7 VRE (4.3%) and 14 VSE (1.6%); blood 203 (19.6%): 54 VRE (32.9%) and 149 VSE (17.1%); CSF 11 (1.1%): 2 VRE (1.2%) and 9 VSE (1%); respiratory samples 49 (4.7%): 3 VRE (1.8%) and 46 VSE (5.3%); Soft tissue 10 (1%): 1 VRE (0.6%) and 9 VSE (1%); urine 521 (50.4%): 69 VRE (42.1%) and 452 VSE (52%); wounds 120 (11.6%): 17 VRE (10.4%) and 103 VSE (11.8%); other sources 39 (3.8%): 3 VRE (1.8%) and 36 VSE (4.1%) (*p* = 0.001). See Table 1, Figure 2.

Patient outcomes showed that 710 (73.3%) survived—85 (56.3%) VRE and 625 (76.5%) VSE—while 258 (26.7%) died—66 (43.7%) VRE and 192 (23.5%) VSE (*p* < 0.001). See Table 1, Figure 3.

The trend of VRE to total study population was 6.1, 7.3, 3.7, 4.9, 5.5, 7.9, 9.1, 17.1, 16.5, and 22 while the trend of VSE to total study population was 4.4, 10.3, 6.2, 6.7, 9.2, 10.2, 10.1, 15.2, 13.1, 14.6 in the years 2012 to 2021, respectively (*p* = 0.02). Trends of *E. faecalis* and *E. faecium* incidence across the years of study are shown in Table 1, Figure 4.

### 3.2. Resistance Patterns against Selected Antimicrobial Agents

This study identified and reported eight different species of Enterococcus: *E. avium*, *E. casseliflavus*, *E. durans*, *E. faecalis*, *E. faecium*, *E. gallinarum*, *E. hirae*, and *E. raffinosus*. *E. faecium* showed high resistance to ampicillin (81.6%), whereas *E. faecalis* showed only 11.8% resistance. Linezolid resistance was observed only in *E. faecium* and *E. faecalis* cases, with prevalence rates of 3.1% and 1.5% in the total sample population, respectively. The highest resistance in *E. faecium* was observed against ciprofloxacin (84.1%), followed by ampicillin (81.6%), rifampicin (80%), and vancomycin (41.2%). The least resistant antibiotics for *E. faecium* were daptomycin (0.6%) and linezolid (3.1%). For *E. faecalis*, the highest resistance was observed against ciprofloxacin (59.7%), followed by rifampicin (20.1%) and ampicillin (11.8%). The least resistant antibiotics for *E. faecalis* were daptomycin (0%), linezolid (1.5%), and vancomycin (2.7%). (Table 2). Trends in ampicillin and vancomycin resistance in the whole study population, ICU, and general ward, as well as trends of *E. faecalis* and *E. faecium* over the study years, are shown in Figure 5. Ampicillin resistance increased in all groups between 2012 and 2021, particularly among ICU patients. No significant change in antibiotic resistance was observed in the last two years of the COVID-19 pandemic; however, if only ICU data is considered, there was a significant increase in ampicillin resistance in 2020 and 2021. Vancomycin resistance in ICU patients decreased slightly between 2020 and 2021, but it increased in the overall population and general ward patients. See Figure 5.

## 4. Discussion

In this study, we examined the prevalence and resistance patterns of *Enterococcus* species in the ICU and general ward of a tertiary care hospital in Saudi Arabia from 2012 to 2021. We identified and reported eight different *Enterococcus* species: *E. avium*, *E. casseliflavus*, *E. durans*, *E. faecalis*, *E. faecium*, *E. gallinarum*, *E. hirae*, and *E. raffinosus*. *E. faecalis* was the most frequently isolated *Enterococcus* species (54.3%), followed by *E. faecium* (33.5%). Our findings support the notion that *E. faecalis* is the most common cause of enterococcal infections [10,11]. However, similar to other studies, our study showed an increasing proportion of infections caused by *E. faecium*. This suggests that *E. faecium* could become the dominant species causing enterococcal infections in the future. Therefore, it is important to monitor the prevalence of *Enterococcus* species over time.

Our findings revealed that VRE constituted 15.9% of the total sample, while VSE made up 84.1%. This VRE incidence was consistent with recent studies [12,13]. The highest incidence of VRE was observed in *E. casseliflavus* (75%), followed by *E. gallinarum* (50%). However, these isolates were few, with only four and six samples respectively, indicating that more positive isolates are needed to represent the true population. These species are known to be intrinsically resistant to vancomycin due to the natural presence of resistant genes [14]. *E. faecalis* represented 54.3% of the total population, with 2.7% of VRE cases, while *E. faecium* represented 33.6% of the total population, with 41.2% of VRE cases. This was similar to a previous report [4], in which 41.7% of ICU isolates exhibited resistance to vancomycin, with 60% of these strains being *E. faecium* and 40% *E. faecalis*. According to our findings, the resistance pattern (VRE) was more prevalent in the ICU population (24.9% of total ICU samples vs. 12.1% of total general ward samples). A high incidence of VRE in ICU samples has also been variably reported in previous studies [4,7,12], with rates of 41.7%, 12.3%, and 17.3%, respectively. This could be because ICU patients are more susceptible to nosocomial infections due to weakened immunity, frequent exposure to antimicrobial agents, and severe illnesses.

In our study, VRE samples in males accounted for 53.7% of total VRE samples. However, the percentage of VRE among total female samples was 19.6%, while it was 13.6% of total male samples. Current literature suggests differences between sexes, with a male preference (59%) in the distribution of VRE [15]. Previous research has found gender differences in infections caused by some pathogens such as Staphylococcus aureus (male predominance) [16] and *Escherichia coli* (female predominance) [17]. Genetic [18] and hormonal factors [18,19] could contribute to this phenomenon.

According to our findings, linezolid resistance was found to be 3.1% and 1.5% in the total sample population in *E. faecalis* and *E. faecium* cases, respectively. Although linezolid resistance is rare, with more than 99% of gram-positive bacteria still susceptible [20], antimicrobial surveillance studies have revealed that the number of linezolid-resistant *Enterococci* has recently increased [21,22,23]. Also, the highest resistance for *E. faecalis* was observed against ciprofloxacin (59.7%), followed by rifampicin (20.1%) and ampicillin (11.8%), and the least resistant antibiotics were daptomycin (0%), linezolid (1.5%), and vancomycin (2.7%). On the other hand, the highest resistance for *E. faecium* was observed against ciprofloxacin (84.1%), followed by levofloxacin (83.6%), penicillin (82.3%), and ampicillin (81.6%), while the least resistant antibiotics were daptomycin (0.6%), tigecycline (2.7%), and linezolid (3.1%). A huge difference in resistance was observed against ampicillin: *E. faecalis* was only 11.8% resistant to ampicillin, whereas *E. faecium* was 81.6% resistant. Ampicillin resistance in *E. faecium* is triggered mainly by increased PBP5 production and/or polymorphisms in the protein’s beta subunit [24].

Another significant difference in resistance was observed against rifampin, with rates of 20.1% and 80% against *E. faecalis* and *E. faecium*, respectively. Previous research on isolates from bovine milk supported our findings, with 13.6% of *E. faecalis* and 42.3% of *E. faecium* being rifampicin-resistant strains [25]. Rifampicin resistance rates were also found to be 71.2% and 94.3% in *E. faecalis* and *E. faecium* isolates from various clinical infections, respectively [26]. Previously, research [27] on urinary tract infection isolates found complete resistance to rifampicin in *E. faecium* (100%) and a significant proportion in *E. faecalis* (81.2%). Rifampicin is not commonly used in the treatment of enterococci as acquired resistance to rifampicin has been observed in both *E. faecium* and *E. faecalis* due to mutations in the gene encoding the RNA polymerase subunit (rpoB) [28,29]. Recently, it was reported that HelD proteins from high G+C *Actinobacteria*, called HelR, were able to dissociate rifampicin-stalled RNA polymerase from DNA and provide rifampicin resistance [25].

Despite the fact that the COVID-19 pandemic was the primary cause of increased and inappropriate antimicrobial use [30], our study found no conclusive evidence of a rising trend of antimicrobial resistance as a result of the pandemic. Other studies have produced conflicting results, with some indicating that the pandemic was associated with an increase in VRE infection or colonization [31] while others discovered a decrease in VRE resistance during the COVID-19 pandemic [4,32]. This disparity could be explained by the fact that some hospitals, possibly due to the heavy load they were exposed to during the pandemic, had decreased infection control activities, decreased adherence to blood culture and central line bundles, and antibiotic abuse in COVID-19 patients, as well as high rates of staff turnover and the presence of non-strictly qualified staff members in ICU settings during the first pandemic periods [33].

Our findings revealed that the mortality rate for VRE infections was higher (43.7%) than for VSE cases (23.5%). However, the effect of VRE infection on mortality is still debatable, with the associated comorbidities potentially skewing the estimates [34,35,36].

Due to the study’s retrospective design, one limitation was that no genomic analysis of resistant isolates, particularly VRE, was performed. Future research involving genomic analysis is warranted. Also, our study represents a regional—or, at most, a national—observation, which may limit its generalizability.

## 5. Conclusions

This study provides an overview of enterococcal infections and antibiotic susceptibility trends, guiding clinicians in the selection of appropriate empirical antibiotic therapy to improve clinical outcomes. From clinical specimens in a tertiary care hospital, we identified eight different *Enterococcus* species. *E. faecalis* was the most commonly identified strain, while *E. faecium* specimens had the highest percentage of VRE. VRE cases had a significantly higher mortality rate than VSE cases. Daptomycin, linezolid, and vancomycin had the least resistance among isolated strains.

## Figures and Tables

**Figure 1 diagnostics-14-01190-f001:**
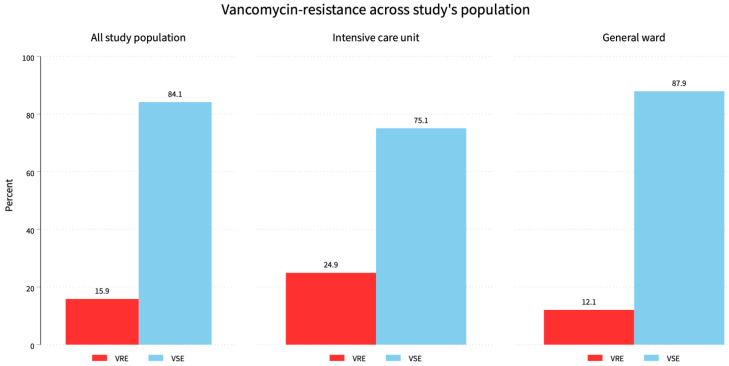
Vancomycin resistance across the all the study’s population. The overall vancomycin resistance was 15.9% while ICU patients had a 46.4% resistance burden and 53.6% for general ward patients.

**Figure 2 diagnostics-14-01190-f002:**
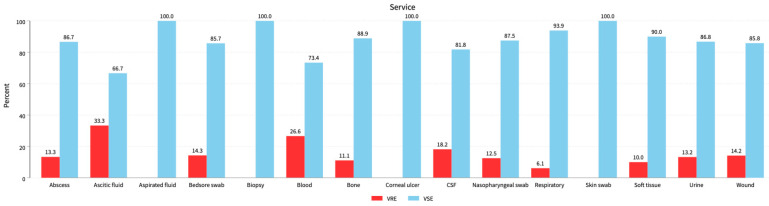
Distribution of VRE isolates among all clinical specimens. Ascitic fluid, blood, and CSF represent the 33.3%, 26.6%, and 18.2% VRE populations followed by bedsore swabs, wounds, and urine samples with 14.3%, 14.2%, and 13.2% VRE populations, respectively.

**Figure 3 diagnostics-14-01190-f003:**
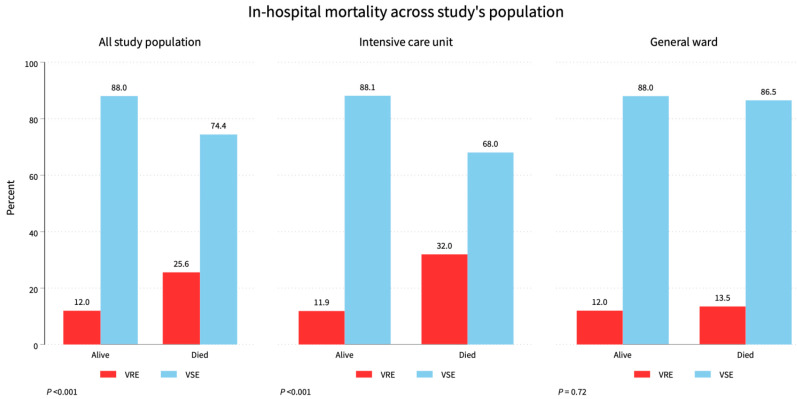
Mortality status across the study population. Total study mortality was 25.6%, while ICU had 32%, and the general ward had a 13.5% mortality rate of VRE.

**Figure 4 diagnostics-14-01190-f004:**
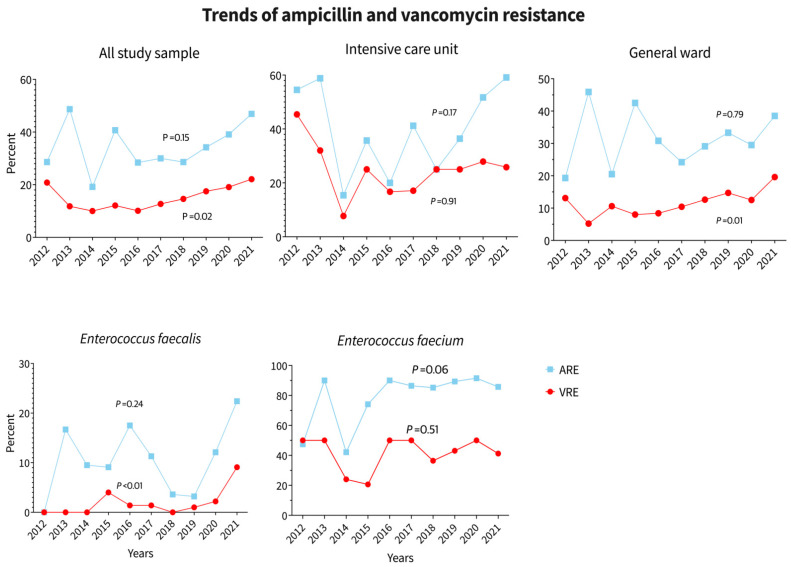
Trends of *E. faecalis* and *E. faecium* incidence across the years of study.

**Figure 5 diagnostics-14-01190-f005:**
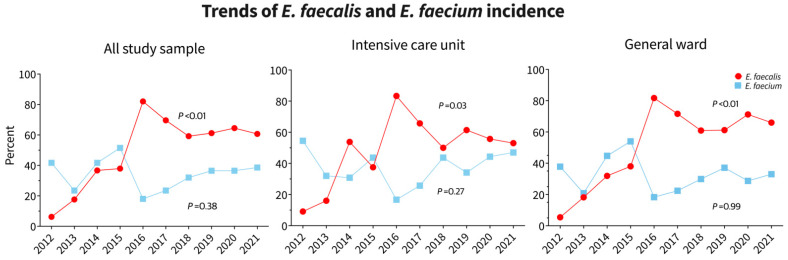
Trends of ampicillin and vancomycin resistance in the whole study population, ICU, and general ward as well as trends in *E. faecalis* and *E. faecium* over the study years.

**Table 1 diagnostics-14-01190-t001:** Characteristics of the study population.

	Total Samples	VRE	VSE	
Characteristic—No. (%)	*n* = 1034	*n* = 164 (15.9)	*n* = 870 (84.1)	*p* Value
Sex				0.01 ^b^
Man	647 (62.6)	88 (53.7)	559 (64.3)	
Woman	387 (37.4)	76 (46.3)	311 (35.7)	
Hospitalization ward				<0.001 ^a^
ICU	305 (29.5)	76 (46.4)	229 (26.3)	
General ward	729 (70.5)	88 (53.6)	641 (73.7)	
*Enterococci* species				<0.001 ^a^
*E. faecalis*	561 (54.3)	15 (9.1)	546 (62.8)	
*E. faecium*	347 (33.6)	143 (87.2)	204 (23.4)	
Other	126 (12.2)	6 (3.7)	120 (13.8)	
Culture source				<0.001 ^a^
Abscess	60 (5.8)	8 (4.9)	52 (6)	
Ascitic fluid	21 (2)	7 (4.3)	14 (1.6)	
Blood	203 (19.6)	54 (32.9)	149 (17.1)	
CSF	11 (1.1)	2 (1.2)	9 (1)	
Respiratory	49 (4.7)	3 (1.8)	46 (5.3)	
Soft tissue	10 (1)	1 (0.6)	9 (1)	
Urine	521 (50.4)	69 (42.1)	452 (52)	
Wound	120 (11.6)	17 (10.4)	103 (11.8)	
Other	39 (3.8)	3 (1.8)	36 (4.1)	
Patient outcome ^c^				<0.001 ^a^
Alive	710 (73.3)	85 (56.3)	625 (76.5)	
Died	258 (26.7)	66 (43.7)	192 (23.5)	
Year				0.02 ^b^
2012	48 (4.6)	10 (6.1)	38 (4.4)	
2013	102 (9.9)	12 (7.3)	90 (10.3)	
2014	60 (5.8)	6 (3.7)	54 (6.2)	
2015	66 (6.4)	8 (4.9)	58 (6.7)	
2016	89 (8.6)	9 (5.5)	80 (9.2)	
2017	102 (9.9)	13 (7.9)	89 (10.2)	
2018	103 (10)	15 (9.1)	88 (10.1)	
2019	160 (15.5)	28 (17.1)	132 (15.2)	
2020	141 (13.6)	27 (16.5)	114 (13.1)	
2021	163 (15.8)	36 (22)	127 (14.6)	

^a^—Fisher’s exact test; ^b^—Binomial generalized linear model for trend; ^c^—Data for 968 unique patients.

**Table 2 diagnostics-14-01190-t002:** Resistance patterns against selected antimicrobial agents of the whole study population.

	Total	Ampicillin	Ciprofloxacin	Daptomycin	Linezolid	Rifampin	Vancomycin
*Enterococci* Species—No. (%)	*n* = 1034	*n* = 939	*n* = 625	*n* = 524	*n* = 913	*n* = 453	*n* = 1034
*E. avium*	1 (0.1)	S	S	S	S	NA	S
*E. casseliflavus*	4 (0.4)	S	S	NA	S	NA	3 (75)
*E. durans*	1 (0.1)	S	NA	NA	S	NA	S
*E. faecalis*	561 (54.3)	62 (11.8)	224 (59.7)	S	8 (1.5)	56 (20.1)	15 (2.7)
*E. faecium*	347 (33.6)	253 (81.6)	174 (84.1)	1 (0.6)	10 (3.1)	124 (80)	143 (41.2)
*E. gallinarum*	6 (0.6)	3 (50)	3 (60)	S	S	2 (50)	3 (50)
*E. hirae*	3 (0.3)	S	NA	NA	S	NA	S
*E. raffinosus*	1 (0.1)	R	S	S	S	S	S
Other	110 (10.6)	21 (23.9)	18 (51.4)	S	S	3 (20)	S

NA—Resistance pattern is not available for the antimicrobial agent, R—All isolated samples are resistant to the antimicrobial agent, S—All isolated samples are sensitive to the antimicrobial agent.

## Data Availability

Data will be available upon reasonable request.

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
