# Peer review of "Characterization and Antimicrobial Susceptibility Patterns of Enterococcus Species Isolated from Nosocomial Infections in a Saudi Tertiary Care Hospital over a Ten-Year Period (2012–2021)"

_diagnostics, 2024, doi:10.3390/diagnostics14111190_

Round 1
Reviewer 1 Report
Comments and Suggestions for Authors
Thank you for the job that you have done.
The authors described the antibiotic resistance profiles among the different Enterococcus Species in one hospital.
It sounds good for the national readers and fills the gaps on the best antibiotic of choice in the study region.
Introduction:
Line 53: Please write the scientific names in Italic format "E. faecalis"
Materials and methods:
About nosocomial infections that you mentioned in title of the article. How did you differentiate the infections as hospital-acquired or community-acquired?
Please be meticulous in the writing of methods. Which standard laboratory SOP was used? Is it CLSI? Please mention it.
The whole section needs to revise as:
Line 81-82: please rewrite the phrase scientifically "Isolates included abscesses, body fluids, blood, CSF, respiratory samples, soft tissue, urine, and wounds." You mean isolates isolated from ……. Cultures”
Results:
Line 105: please check the writing, for example "(P0.001)" has to be "(P=0.001).
Line 120: about the sentence “we used following culture sources…” I think it better to say “distribution of different resistance profiles among isolates from different sources were…..” You just did the data collection.
Line 132-133: unclear sentence for me (writing format)
Discussion:
It is too wordy. Can be shorter with citations to other studies.
Line 200-205: it seems to be your speculation about gender-related infection for the bacterium. Please avoid it or support it with scientific facts.
Comments on the Quality of English LanguagePlease revise the writing. punctuations are not correct.
Author Response
Dear reviewer
We would like to sincerely thank you, the editorial office and the reviewers for the time and effort taken to consider our manuscript. All the comments we received are highly appreciated and fully considered by us. Detailed response is in table below.
Sincerely,
Authors

Reviewer 2 Report
Comments and Suggestions for Authors
- This is original type; Thus, abstract should structured e.g., introduction, method, results, as well as conclusion.
- Abstract length was too much.
- the importance of the work should be better highlighted in the introduction.
- Exclusion and inclusion criteria should be added to methods.
- The bacterial genus/species should be stated with italic font through the text.
- The author should used Odds ratio/Crud Odds ratio between VRE infection and patients characteristics particularly poor outcomes e.g., mortality risk.
- Discuss about novelty of the work than previous published studies.
- Discuss about study limitation.
Comments on the Quality of English LanguageMinor language polishing was needed on this manuscript.
Author Response

(The authors gave the same response as above.)

Reviewer 3 Report
Comments and Suggestions for Authors
This study investigated changes in the species distribution and antibiotic susceptibility of Enterococcus spp. isolated over a 10-year period from a single university hospital. Despite the title suggesting analysis of Enterococci isolated from nosocomial infections, authors does not describe how nosocomial infections were defined in the method section. The distribution and resistance profiles of the strains suggest a likelihood of inclusion of community-acquired strains. Furthermore, it is unclear whether both colonizing and infecting strains were included, or only strains from infected patients. Given the emphasis on the epidemiological distribution of Enterococci isolates in this study, it seems inappropriate to show a relationship between VRE and patient mortality. Instead, a comparison of species distribution and antibiotic susceptibility of Enterococci isolated from each specimen seems more clinically meaningful. Additionally, considering the abundance of epidemiological studies on Enterococci, the clinical significance of a simple epidemiological investigation of isolates from a single university hospital appears limited.
Author Response

(The authors gave the same response as above.)

Round 2
Reviewer 1 Report
Comments and Suggestions for Authors
Good job
Comments on the Quality of English Language
Minor revise is necessary
Reviewer 3 Report
Comments and Suggestions for Authors
The authors seem to have revised the paper appropriately according to the reviewer's questions.